# Emerging Insights into the Function of Kinesin-8 Proteins in Microtubule Length Regulation

**DOI:** 10.3390/biom9010001

**Published:** 2018-12-20

**Authors:** Sanjay Shrestha, Mark Hazelbaker, Amber L. Yount, Claire E. Walczak

**Affiliations:** 1Medical Sciences Program, Indiana University, Bloomington, IN 47405, USA; sashrest@indiana.edu (S.S.); maanhaze@umail.iu.edu (M.H.); 2Department of Molecular and Cellular Biochemistry, Indiana University, Bloomington, IN 47405, USA; AYount@franklincollege.edu

**Keywords:** microtubule dynamics, mitosis, spindle, molecular motor protein

## Abstract

Proper regulation of microtubules (MTs) is critical for the execution of diverse cellular processes, including mitotic spindle assembly and chromosome segregation. There are a multitude of cellular factors that regulate the dynamicity of MTs and play critical roles in mitosis. Members of the Kinesin-8 family of motor proteins act as MT-destabilizing factors to control MT length in a spatially and temporally regulated manner. In this review, we focus on recent advances in our understanding of the structure and function of the Kinesin-8 motor domain, and the emerging contributions of the C-terminal tail of Kinesin-8 proteins to regulate motor activity and localization.

## 1. Microtubules and Dynamic Properties

### 1.1. Microtubule Structure

Microtubules (MTs) are dynamic polymers composed of α- and β-tubulin heterodimers. These heterodimers polymerize into protofilaments in head-to-tail arrangements, and 13 of these protofilaments associate together through lateral connections to form a hollow tube [1]. Because the individual subunits are heterodimers, there is a structural polarity of the MT with α-tubulin at the minus end and β-tubulin at the plus end [2]. Despite the similarities of α- and β-tubulin in structure and GTP-binding ability, only β-tubulin is capable of hydrolyzing GTP [3]. GTP-bound heterodimers are preferentially added to the plus ends of growing MTs during polymerization and then GTP is hydrolyzed, such that the lattice is predominately composed of GDP-tubulin. A lag in the rate of hydrolysis as compared to polymerization generates a GTP-tubulin cap, which is thought to stabilize the MT [4,5]. The size of the GTP cap and its effects on the structure and dynamics of MTs are areas of active study [5,6,7,8,9,10,11,12].

Structurally, tubulin heterodimers in the MT lattice are straight because they are constrained by lateral protofilament interactions. At growing MT ends, the MTs are thought to grow as sheets that then close. Most models postulate that when a MT transitions from growth to shrinkage, the protofilaments bend outwards with a higher degree of curvature [13,14,15]. However, a recent elegant electron tomography study showed that the curvature of protofilaments is similar on growing and shortening MTs both in vitro and in vivo, suggesting that GTP-bound tubulin is bent and only straightened when incorporated into the MT lattice [12].

### 1.2. Dynamic Instability

Microtubules demonstrate a unique behavior termed dynamic instability in which they co-exist in states of growth (polymerization) and shrinkage (depolymerization) and interconvert stochastically between these two states [5]. Microtubules can undergo a switch from growth to shrinkage, called a catastrophe, or a switch from shrinkage to growth, called a rescue [16]. These dynamic properties are also linked to the MT structure. One model to explain MT catastrophes posits that loss of the GTP-cap releases stored energy from the constrained tubulin dimers in the MT lattice, resulting in a catastrophe [5]. Consistent with this model, recent high resolution cryo-electron microscopy (Cryo-EM) studies suggest that the release of phosphate upon GTP hydrolysis from β-tubulin causes α-tubulin compaction, inducing strain in the MT lattice, which is released upon depolymerization [17]. A second, albeit not mutually exclusive, model stems from the observation that the rate of catastrophe can be correlated with MT age in vitro [9,18]. In this model, accumulation of defects like staggering protofilament growth or lattice defects, which are thought to be associated with MT aging, promotes catastrophe [18]. Microtubule rescues are more poorly understood, but recent work suggests that MT severing enzymes may play a unique role in MT rescue by creating lattice damage that is repaired by incorporation of GTP-tubulin [19].

Microtubules in vivo are known to be more dynamic than in vitro. These dynamics allow cells to remodel their cytoskeleton for various purposes. For example, MTs exhibit a dramatic increase in dynamic instability when cells enter mitosis, which is governed largely by an increase in the frequency of catastrophe [20,21,22]. This allows for a global reorganization of the MTs to form the mitotic spindle, which separates chromosomes in a dividing cell. The mitotic spindle consists of functionally distinct classes of MTs, each with different dynamic properties. Spindle MTs (interpolar) extend from the two poles toward the spindle midzone, contribute to the polar ejection force [23] and maintain the general architecture of the mitotic spindle. Spindle MTs have a half-life of ~10 s [24]. Kinetochore MTs connect the spindle poles to kinetochores; they have a longer half-life of ~2.5 min [25]. Astral MTs extend from the poles toward the cell periphery and help define spindle positioning in diverse organisms [26,27]. Their half-life is similar to the half-life of spindle MTs [24]. How the spindle maintains these diverse populations of dynamic MTs is an active area of investigation.

The dramatic changes in MT dynamics that occur in vivo rely on a variety of MT associated proteins (MAPs) to regulate MT dynamics. These MAPs include MT stabilizing proteins that bind along the length of the MT lattice [28] and MT destabilizing proteins that alter MT end structure or that sever the MT lattice. Proper MT function during mitosis requires a host of both stabilizing and destabilizing proteins to properly regulate spindle structure [29]. Of particular interest are destabilizers that act on MT ends, including motor proteins in the Kinesin-13 and Kinesin-8 families. In this review, we focus on recent advances in our understanding of Kinesin-8 family members in controlling the dynamic properties of MTs.

## 2. Kinesin-8 Family of Motor Proteins

Kinesin-8 proteins are members of the kinesin superfamily of molecular motors that utilize ATP hydrolysis for movement along MTs. Kinesin-8 proteins are MT plus-end directed motors and plus-end MT destabilizing enzymes. While some organisms have a single Kinesin-8 motor protein, others have multiple Kinesin-8 family members (Table 1) that act on different subsets of MTs to execute both mitotic and non-mitotic functions [30].

### 2.1. Localization

In cells, Kinesin-8 proteins localize primarily to the growing plus ends of MTs [31,32,33,34,35,36,37]. Kinesin-8 proteins accumulate to different levels on subsets of MTs such that not all populations of MTs have the same amount associated with them [33,34,35]. Even on a single population of MTs, Kinesin-8 proteins have been observed to be localized in a gradient, such that the intensity varies along the length of the MTs [38,39]. It should be noted that not all Kinesin-8 proteins are localized via movement on MTs, as *Drosophila* Klp67A localizes at the kinetochore, independent of MTs [40].

### 2.2. Cellular Roles

Kinesin-8 proteins are involved in a variety of cellular processes where they control MT length. Early studies in budding yeast revealed a role for Kip3 in nuclear positioning [41,42,43,44]. Consistent with this functional role, in higher eukaryotes, Kif18B was shown to play a role in the proper orientation and positioning of the metaphase spindle [45]. A large number of Kinesin-8 proteins have been shown to regulate spindle length in many organisms [32,36,38,40,46,47,48,49,50,51,52,53], which may also contribute to their roles in chromosome dynamics. However, not all Kinesin-8 proteins function solely in MT length regulation, as some Kinesin-8 proteins also contribute to MT cross-linking and sliding [47,54].

Kinesin-8 proteins also play essential roles during development and in disease. Knockout of Kif18A in mice disrupts testis development, resulting in sterility [55]. Consistent with these studies, a missense mutation in a highly conserved residue in the Kif18A motor domain resulted in cell cycle arrest and apoptosis of germ cells during embryogenesis, leading to infertility in both sexes [56]. In mice, Kif19A controls the proper length of motile cilia, which contributes to fluid flow in various tissues [37]. In support of this function, Kif19A null mice had elongated cilia in neuronal, tracheal, and oviduct epithelial cells, which manifested in hydrocephalus and female infertility [37]. The Kinesin-8 proteins also may be important in cancer. For example, Kif18A is mis-expressed in numerous cancers, which correlates with advanced tumor grade and poor survival [57,58,59,60,61,62,63,64], whereas Kif18B has been implicated in tumor progression through Wnt signaling and may be a driver in carcinogenesis [65,66]. Together these studies highlight the diversity of functions involving Kinesin-8 proteins.

## 3. Biophysical Properties of Kinesin-8 Proteins

Kinesin-8 proteins have an N-terminal motor domain, followed by a class-specific neck, a stalk domain that allows dimerization, and a C-terminal tail that is utilized for cargo binding. In most Kinesin-8 proteins, the tail has an additional MT binding site that allows the kinesin to tether to the MT lattice [34,49,67,68,69]. However, some Kinesin-8 tail domains have binding sites for regulatory proteins, such as End-binding Protein 1 (EB1), importin-α, and Mitotic Centromere-Associated Kinesin (MCAK), which control localization and activity of the motor protein [34,35].

Kinesin-8 proteins are MT plus-end directed motors and plus-end MT destabilizing enzymes (Table 2). Most Kinesin-8 proteins are fairly slow motors (~3 µm/min), and many are highly processive, traveling ~10 µm before dissociating from the MT [31,68,70]. Processivity allows motor proteins to travel long distances in cells, and for Kinesin-8 proteins it may be important to help them reach the ends of MTs in the spindle. Kif18A is approximately 5-fold faster than the other Kinesin-8 proteins, while still maintaining high processivity [68,70]. Not all Kinesin-8 proteins are highly processive, as Kif18B was initially reported to be much less processive due to switching of the motor between a diffusive state and directed motility until it reached the MT end, where it dwells at the MT end [71]. However, a more recent study showed that Kif18B was highly processive, and that its motility and processivity required the C-terminal tail but not its interaction with EB1 [45]. The differences between these studies on Kif18B and how processivity contributes to Kinesin-8 function are important avenues for future investigation.

A key feature of the Kinesin-8 proteins is that they are MT destabilizers, but they appear to use different mechanisms to achieve this activity (Figure 1). Early studies with yeast showed that Kip3 could depolymerize stabilized MT substrates [31,33]. Interestingly, Kip3 was more active on longer MTs, which is postulated to be a result of a concentration gradient of Kip3 near the MT plus ends that may form due to its high processivity and low off rate from MT ends [72]. The ability of Kip3 to depolymerize stabilized MT substrates may be due to its tight binding to a curved conformation of tubulin at the end of a MT [76]. Early studies showed that Kif18A could also depolymerize stabilized MT substrates [32], but later studies proposed an alternative mechanism for destabilization (Section 4.3) [78]. Kif19A can depolymerize stabilized MT substrates [37], and it also has the ability to interact with and stabilize curved MT substrates [79], suggesting that MT end structure may be an important component of the ability of Kinesin-8 proteins to directly depolymerize MTs.

Other members of the Kinesin-8 family have been shown to destabilize dynamic MTs but not actively depolymerize stabilized MTs in vitro. For example, *S. pombe* Klp5/6 facilitates MT depolymerization in vivo [50,52,80], but it was not able to directly depolymerize stabilized MTs in vitro [74]. Similarly, while Kif18B could dwell at the plus ends of dynamic MTs, it was unable to actively depolymerize stabilized MTs [45,71]. Du and colleagues showed that Kif18A could cap the plus end of a MT, preventing both polymerization and depolymerization [78] (Figure 1B), ultimately leading to a MT catastrophe. This is consistent with the observation that Kif18A promotes MT pausing in a concentration-dependent manner [81]. How Kif18A and other Kinesin-8 motor proteins affect all parameters of MT dynamic instability will be an important avenue of future investigations.

One current area of study is to elucidate the functional domains of Kinesin-8 proteins that contribute to their MT destabilization activity. For example, several monomeric Kinesin-8 proteins containing only the motor domain can depolymerize MTs, although there is a compromise in MT depolymerization activity when compared to dimeric full-length proteins [76,77,79], which may be due to dimerization or to additional domains present in the full-length protein. In contrast, a dimeric version of Kif18B, containing the motor, neck and stalk, but lacking the tail, is able to depolymerize stabilized MTs, whereas full-length constructs of Kif18B containing the tail cannot, suggesting that the tail of Kif18B may actually be inhibitory to MT depolymerization [45].

Together, these studies suggest that although Kinesin-8 proteins share a common function of negatively controlling MT length, different mechanisms underlie this task. Perhaps subtle differences within the motor domains could be sufficient to confer diversity in the biophysical properties that ultimately dictate how Kinesin-8 proteins destabilize MTs. Alternatively, it may be that the non-motor tail domains are critical to localization and function, which may help modulate their activities in cells.

## 4. Emerging Insights into the Structure and Function of Kinesin-8 Motor Domain

Current models propose that dimeric Kinesin-8 proteins walk along the MT toward the plus end where they destabilize MTs; thus, motility and destabilization appear to be intimately associated. However, the detailed molecular mechanisms of the relationship between Kinesin-8 motor protein motility and MT destabilizing activity are poorly understood. Recent studies have found that both motile and non-motile monomeric motor domains can depolymerize MTs from both ends in vitro, suggesting that motility and destabilization are two independent activities of Kinesin-8 proteins. These studies also suggest that MT destabilization activity is an intrinsic property of the motor domain [76,77,79]. Further support for the importance of the motor domain comes from structural studies of monomeric Kif18A and Kif19A proteins coupled with domain swap experiments, which have provided new insights on critical elements of the Kinesin-8 motor domain that contribute to both motility and MT depolymerization.

### 4.1. Insights from Structural Studies

#### 4.1.1. Kif18A

The motor domain of Kif18A bound to Mg^2+^-ADP has the canonical arrowhead shape with a central β-sheet at the core and three α-helices on either side (Figure 2A) [82,83,84]. The regions around the nucleotide binding pocket are also largely similar to other kinesin structures, but loop L11, which coordinates nucleotide binding with MT binding, is disordered [82]. The key MT binding element, helix α4, is positioned similarly to in other kinesins bound to ATP analogs. On the other hand, in addition to loop L11, loops L8 and L12, which also have conserved roles in MT binding [85], are also partially disordered in the crystal structure [82]. Likewise, loop L2, which is critical for the MT depolymerization activity of Kinesin-13 proteins [86,87], is flexible and is not fully visible in the crystal structure [82].

A cryo-EM reconstruction of the nucleotide-free Kif18A motor domain and neck linker region docked onto a straight MT revealed several key insights into how Kinesin-8 proteins change structure upon MT binding. The α4 relay helix is positioned at the tubulin intradimer interface and interacts mainly with helix H12 of α-tubulin, similar to what has been found with other kinesins (Figure 2B) [82]. The α4 helix becomes extended upon MT binding consistent with changes found in several other kinesin superfamily members [77,88,89,90,91]. In addition, helix α6 moves closer to helix H12 of α-tubulin toward the MT minus end, and loop L8 becomes more structured and points in the opposite direction toward helix H12 of β-tubulin at the plus end of the MT, which would facilitate multiple MT binding interactions. Loop L2 becomes slightly more ordered and appears to point towards helices H5 and H12 of α-tubulin at the minus end of the MT [82].

Upon MT binding, conformational changes of the Kif18A α4 relay helix open the nucleotide binding pocket, which may help accelerate ADP release and facilitate ATP binding [77,82,92]. After ATP binds (AMP-PNP bound structure, PDB: 5OCU), the conserved structural elements near the nucleotide binding pocket, the P-loop, switch I (L9) and switch II (L11), undergo structural reorganization closing the nucleotide binding site [77]. These conformational changes at the nucleotide binding site lead to the opening of the neck linker docking cleft and induce an extension of helix α6, resulting in the reorientation and docking of the neck linker sequence towards the plus end of the MT [77,89]. In vitro biochemical data suggest that neck linker docking greatly enhances the ATPase activity, MT affinity, motility and depolymerization activity of the motor domain [77].

#### 4.1.2. Kif19A

Recent structural studies revealed that the murine Kif19A motor domain plus neck linker in the Mg^2+^-ADP state also has the arrowhead structure typical of the kinesin superfamily [79]. However, unlike what has been shown for other kinesins, there is a dramatic shift of the switch II cluster (α4-L12-α5), which extends away from the catalytic core and is less ordered than in most kinesin structures (Figure 2C). The α4 helix, which is the major MT binding element, is more distant from the central core of the motor relative to other kinesins. Other notable differences include a longer loop L2, a flexible loop L8, and a shorter α6 helix. Loop L2 is more ordered in the Kif19A structure compared to the Kif18A structure, and it points in the direction of the MT minus end. Loop L8, which normally points toward the plus end of the MT, is more retracted toward the catalytic core [79].

A cryo-EM image reconstruction of the nucleotide-free Kif19A motor plus neck linker region docked onto the MT illustrates dramatic structural changes that occur upon Kif19A binding to MTs (Figure 2D). Notably, the switch II cluster retracts toward the motor domain core compared to its highly extended position in the unbound structure, which permits the interaction of helix α4 with helix H12 of α-tubulin at the tubulin intradimer interface, consistent with what was seen with Kif18A. Similar to Kif18A, binding of the Kif19A motor domain to the MT lattice leads to structural changes in the motor domain that would accelerate ADP release from the active site and rotates loop L8 so that it can interact with helix H12 of β-tubulin. In contrast, loop L2 does not undergo marked movement upon MT binding, but it extends to the minus end of the tubulin interdimer groove toward the H11 helix of the next β-tubulin. There is also a key interaction of both the basic and hydrophobic clusters of loop L2 of Kif19A with the H8-S7 loop and helix H12 of α-tubulin [79].

### 4.2. Key Elements Associated with Microtubule Destabilization

A central focus of current research is to understand which structural features of the motor domain are important for MT destabilization. Previous structural and biochemical studies of the Kinesin-13 proteins, including Kif2C, revealed that loop L2 plays a critical role in MT depolymerization [86,93,94]. In Kinesin-13 proteins, loop L2 consists of two antiparallel β-sheets connected by a highly conserved KVD motif (Lys, Val, and Asp) (Figure 3), which forms a rigid finger-like projection that interacts with α-tubulin at the ends of MTs. This projection is thought to stabilize the intradimer tubulin curvature, resulting in protofilament peeling and MT depolymerization [86,93,94]. In Kinesin-8 proteins, L2 loops are of variable length, but are generally longer than in Kinesin-13; notably, they do not contain the KVD motif. To test whether loop L2 contributes to MT destabilization in Kinesin-8 proteins, domain swap experiments were performed wherein loop L2 from Kif19A was transferred to Kif18A or to Kif5C (a Kinesin-1), and the resulting chimeras were tested for MT destabilization activity. The Kif19A loop L2 slightly increased the MT depolymerization activity of Kif18A but was not able to confer MT depolymerization activity to Kif5C, suggesting that loop L2 alone is not sufficient for MT depolymerization activity [79]. While loop L2 of Kif19A does not have the conserved KVD motif found in Kinesin-13 proteins, it does have a series of acidic-hydrophobic-basic residues between two antiparallel β-sheets, which could act like the KVD finger of Kinesin-13 depolymerases. Mutational studies of loop L2 in Kif19A revealed that a hydrophobic residue (L55) and four basic residues (R56, H58, R59, R61) are critical for MT depolymerization [79]. However, mutation of these residues also lowered the tubulin stimulated ATPase activity and impaired motility, suggesting that their effects on MT depolymerization could be due in part to a reduction of the amount of motor arriving at MT ends. In contrast to the studies of Kif19A, loop L2 of Kip3 is not required for MT depolymerization activity [76]. However, it is required for Kip3 processivity and plus end dwell time, suggesting that it is important for motor function [76]. In human cultured cells, mutational studies of loop L2 of Kif18A revealed that the basic residues of loop L2 are required for localization at the ends of kinetochore MTs, suggesting that loop L2 may contribute either to motility or to MT plus end dwell time [95]. Overall, these studies suggest that loop L2 is not a conserved element essential for MT destabilization activity in the Kinesin-8 family, but it does affect different aspects of the motor and/or MT destabilization activity.

Loop L11 is highly conserved among Kinesin-8 family members and is involved in MT binding. In budding yeast, when loop L11 of Kip3 was replaced by loop L11 of Kinesin-1, the chimeric motor failed to dwell at the MT plus end, and MT depolymerization activity was severely impaired [76]. Furthermore, additional domain swap experiments revealed that while loop L11 is necessary for the MT depolymerization activity of Kip3, the depolymerization activity is also affected by the presence of the neck linker and loop L2 because the presence of these domains significantly improves the dwell time and processivity of Kip3 [76]. The importance of loop L11 in other Kinesin-8 proteins has not been investigated.

Like loop L11 in Kip3, loop L12 of Kif19A was also found to be important for MT binding and for MT depolymerization activity. Specifically, mutational and biochemical studies showed that the basic residues (K290, K294) of Kif19A loop L12 contribute to MT binding and MT depolymerization via electrostatic interactions with the E-hook of β-tubulin [79]. A unique asparagine (N297) in loop L12 of Kif19A confers a higher degree of flexibility to its switch II region, allowing it to bind to both straight and curved MTs. The unique nature of the N297 residue makes it unlikely that other Kinesin-8 proteins depend on loop L12 for MT depolymerization, although this remains to be tested.

Taken together, these studies indicate that, unlike the Kinesin-13 proteins, which have clearly defined regions that contribute to MT depolymerization activity, in the Kinesin-8 family, there is no single element within the Kinesin-8 motor domain that is sufficient for MT depolymerization activity. These studies support the idea that the Kinesin-8 proteins either use a diversity of mechanisms to destabilize MTs or that there are unique structural interactions with the MT that are conserved within the Kinesin-8 family that are not sequence specific.

### 4.3. Emerging Microtubule Depolymerization Mechanisms of Kinesin-8 Proteins

While all Kinesin-8 proteins are known to regulate MT dynamics, it is becoming clear that they do not act via the same mechanism. The structural and biochemical studies described above have provided new insights into how different elements of the motor domain contribute to destabilization at the plus end of MTs. In yeast, Kip3 is proposed to walk along MTs until it reaches a MT end, where ATPase activity is suppressed to halt motility and cause a switch to a high affinity MT binding state with the curved tubulin at the MT plus ends. This tight binding stabilizes the protofilament curvature, ultimately leading to MT depolymerization [76]. Therefore, Kinesin-8 proteins appear to tightly bind to MT ends but do not require ATP hydrolysis for catalytic MT depolymerization. This conclusion is supported through mutational analysis of a conserved residue (E345) that is required for ATP hydrolysis across all kinesins, which did not abolish the ability of the Kip3 motor domain to depolymerize MTs [76]. Additional studies revealed that the Kip3 switch to high affinity binding at the MT plus end is mediated by loop L11 in association with a specific aspartic acid residue (D118) in α-tubulin. Mutation of D118 in α-tubulin abolished the ability of Kip3 to dwell at MT plus ends and resulted in the loss of MT depolymerization activity, suggesting that the interaction of Kip3 loop L11 with the curved tubulin dimer is critical for Kip3-mediated MT depolymerization [76]. Notably, the D118 mutation in α-tubulin did not affect the ability of Kinesin-13 proteins to depolymerize MTs [76]. Additionally, while ATP-bound Kinesin-13 motor proteins are proposed to induce curvature of tubulin dimers at MT ends [96], the binding-switch model proposes that Kip3 instead stabilizes the pre-existing curvature of tubulin dimers through ATP-dependent tight binding leading to MT depolymerization, supporting the idea that Kinesin-8 proteins depolymerize MTs by mechanisms distinct from those utilized by Kinesin-13 proteins.

The mechanisms that other Kinesin-8 family members utilize for MT destabilization are less well defined. There is some evidence suggesting that human Kif18A could depolymerize MTs similarly to Kip3. Consistent with findings for Kip3, loop L2 of Kif18A does not seem to be the primary element needed for MT depolymerization, but it is critical for localization of Kif18A at kinetochore MT plus ends [95], suggesting that similar to Kip3, loop L2 could promote processivity and plus end dwell time for Kif18A. Another conserved mechanism is that neither Kif18A nor Kip3 require ATP hydrolysis for MT depolymerization, suggesting that it is the interaction of the motor with the MT end that is critical rather than the motility [76,77]. In addition, both studies found that the presence of a neck linker greatly improves the MT depolymerization activity of Kif18A and Kip3, and that both Kif18A and Kip3 monomeric motor domains can depolymerize MTs from both ends, suggesting a common mechanism may be involved in depolymerization from either end of the MT [76,77]. Given that there is some debate about the ability of Kif18A to depolymerize MTs, it is possible that a concentration threshold of Kif18A may be required to induce or stabilize curvature for MT depolymerization. Below this threshold, Kif18A could alternatively stabilize the curved end and act as a pausing/capping factor [77].

Structural studies of Kif19A suggest that the dual function of motility and MT depolymerization depends on the conformational flexibility of loop L8 and loop L12 that allow Kif19A to bind both the MT lattice and the curved tubulin at the MT plus end [79]. Like Kinesin-13 proteins, loop L2 plays a central role in MT depolymerization by stabilizing the inter-tubulin dimer interface at the plus end of the MT. In silico studies revealed that the Kif19A motor domain bound to ADP favors curved tubulin binding, which contrasts with both Kip3 and the Kinesin-13 proteins. These studies suggest that different Kinesin-8 proteins have evolved different mechanisms to destabilize MTs.

## 5. Importance of C-Terminal tails in Modulating Kinesin-8 Function

The tail domains of kinesin superfamily members are largely involved in cargo binding, subcellular localization, and modulation of activity [97,98]. Most Kinesin-8 proteins have a second MT binding site in their C-terminal tail that helps tether it to the MT, which contributes to plus-end accumulation, MT destabilization and stabilization, as well as MT crosslinking and sliding. In some instances, the Kinesin-8 tail can interact with other proteins, thereby increasing the diversity of how localization and activity can be controlled [34,35].

### 5.1. Localization and Destabilization Activity

A major function of the C-terminal tails of Kinesin-8 proteins is to control the localization of the motors to MT plus ends. For example, Kip3 requires its tail to accumulate at the plus ends of MTs in vivo [67]. In human cells, tailless Kif18A does not localize at the plus ends of kinetochore MTs, but instead robustly localizes along the length of spindle MTs [49,68,69]. Likewise, the tail of Kif18B is important for localization to the plus ends of astral MTs [34,35]. Intriguingly, the Kinesin-8 C-terminal tails may facilitate more than generic MT binding, as they may also specify the subpopulation of MTs where they enrich. For example, a chimeric protein containing the motor domain of Kif18B fused to the tail of Kif18A localized to plus ends of kinetochore MTs and was able to execute the functions of Kif18A in chromosome alignment [95]. In a reciprocal experiment, a chimeric construct containing the motor domain and neck of Kif18A fused to the tail of Kif18B localized at the plus ends of astral MTs [45]. These studies demonstrate that the Kinesin-8 tails are important in controlling where these proteins act.

Several biochemical studies have indicated that the Kinesin-8 tails also control motor biophysical properties in vitro. For example, tailless Kip3 constructs have a decreased run length and plus-end dwell time. Consistent with this idea, a chimera of the Kip3 tail fused to the motor domain of Kinesin-1 caused the chimeric protein to behave like Kip3 by increasing its processivity and accumulation at MT plus ends [67]. Likewise, in vitro studies of human Kif18A showed that the tail is required for its processivity and plus-end dwell time [68]. The role of the tail of Kif18B is still under debate. One study showed that the MT binding site in the tail of Kif18B contributed to weakly processive diffusion along the MT lattice [71], whereas a more recent study suggests that the tail imparts high processivity and plus-end dwell time [45]. Together, these findings suggest that the tail may modulate motility along the MT lattice for some Kinesin-8 proteins, but that the tail appears to play a conserved role in modulating MT plus end accumulation.

The differential affinity of the Kinesin-8 tails to MTs can manifest in interesting physiological regulation in cells. For example, in yeast, Kip3 has both stabilizing and destabilizing effects on MTs [33,67], which are mediated by the tail. It was shown in vitro and in vivo that the Kip3 tail can inhibit MT shrinkage and stabilize shrinking MTs [39,67]. In support of this idea, a Kinesin-1 construct with the Kip3 tail can stabilize MTs in vivo [67]. Additional biochemical studies revealed that the tail has a more significant effect on modulating MT end affinity rather than lattice affinity, leading to a model wherein higher concentrations of Kip3 at the plus end destabilize the MT ends, whereas lower concentrations of Kip3 slow shrinkage and promote MT rescue [67]. This dual activity is not limited to Kip3, as both *Drosophila* Klp67A and human Kif18A have both stabilizing and destabilizing effects on MTs [38,99].

### 5.2. Crosslinking and Sliding

Not all cellular effects of Kinesin-8 proteins are based solely on their ability to regulate MT dynamics. In budding yeast, the tail of Kip3 is also critical for the crosslinking and sliding of MTs [47,54]. On antiparallel MTs, the Kip3 motor domain walks toward the plus end of one MT while the tail remains relatively stationary on the other, resulting in sliding of the MTs. Conversely, when MTs are in a parallel orientation, Kip3 crosslinks MTs and produces tug-of war movements wherein the sliding of one MT relative to the other is limited due to binding of motors with opposite head-to-tail orientations [47]. In pre-anaphase cells, a delicate balance between cross-linking/sliding and destabilizing activities appears to be critical for maintaining the spindle length. During anaphase, the increase in MT length and decrease in MT overlap allow the Kip3 destabilizing activity to dominate over its crosslinking/sliding activity to regulate post anaphase spindle length [47]. Consistent with this idea, studies in *Drosophila* also revealed that depletion of KLP67A caused spindle defects commonly associated with MT sliding [99,100].

### 5.3. Regulation of Kinesin-8 proteins through Association with Other Regulators

The Kinesin-8 C-terminal tail has binding sites for proteins other than the MT, with the most important regulator being the plus-tip tracking protein EB1 [34,35]. In cells, EB1 is required for robust targeting of Kif18B to the plus ends of MTs [34,35]. The tail of Kif18B also interacts with the Kinesin-13 MCAK, and it has been proposed that a major role of Kif18B is to target MCAK to MT plus ends [35]. While this study showed that Kif18B and MCAK mutually depend on each other for localization at the plus ends of astral MTs, in other systems, knockdown of Kif18B did not perturb MCAK localization [53]. It was also reported that the tail of Kif18B can bind importin-α [34], which may be needed simply to import Kif18B into the nucleus. However, because the importins have been shown to play roles in spindle assembly independent of nuclear transport [101], this finding also raises the possibility that Kif18B could be an additional factor that is spatially regulated by the RanGTP gradient. Finally, the tail of Kif18B was found to be phosphorylated in vivo [102], and phosphorylation of specific residues regulated the affinity of the tail for MTs in vitro [45]. Taken together, these studies illustrate the complex ways in which the tail specifies where and how Kinesin-8 proteins modulate MT dynamics and highlight the need to focus on the underlying regulatory mechanisms that control Kinesin-8 function.

## 6. Conclusions and Future Directions

Kinesin-8 family members play essential roles in MT length regulation. However, the detailed molecular mechanisms by which Kinesin-8 proteins execute this task are still elusive. Recent structural and biochemical studies have provided insights into how the concerted action of motor domain elements confers MT destabilization activity to regulate plus-end dynamics. Kinesin-8 proteins can either act as capping factors, or they can actively depolymerize MTs via induction/stabilization of tubulin curvature. Future studies should be focused on whether these mechanisms are truly distinct or whether capping may ultimately lead to an altered MT structure, resulting in MT depolymerization.

Since budding yeast does not have a Kinesin-13 motor protein and only has a single Kinesin-8 motor, Kip3 may behave differently than other Kinesin-8 proteins. Given the extensive biochemical analyses that have been done with Kip3, future structural studies of Kip3 would be pivotal in allowing direct comparison of its mechanisms to the rest of the Kinesin-8 family. Furthermore, comprehensive biochemical characterization involving domain swaps and mutational analysis of motor head structural elements in other Kinesin-8 proteins are needed to evaluate the key elements involved in MT depolymerization.

While the motor domain is critical for motility and MT destabilization, the C-terminal tail dictates where and how Kinesin-8 proteins act. For example, Kip3 uses its tail to multitask the myriad of activities that it is involved in. The tail of Kip3 controls both MT stabilization and destabilization activity, can spatially and temporally regulate MTs, and crosslinks and slides MTs to maintain spindle integrity [39,47,103]. While other members of the Kinesin-8 family also contain a second MT binding site in the C-terminus, it is not known whether the tails of other Kinesin-8 proteins confer these activities. There is some evidence that Kif18A/Kif18B appear to play a role in the spatial regulation of MTs; however, the mechanism is still unknown [53,81]. It will be critical to dissect the functional interactions mediated by the tail to help reconcile how the tail domains modulate localization and function. Finally, many Kinesin-8 family members contain protein binding motifs and post-translation modification sites whose roles need further investigation in order to determine how Kinesin-8 proteins are modulated both spatially and temporally in vivo.

## Figures and Tables

**Figure 1 biomolecules-09-00001-f001:**
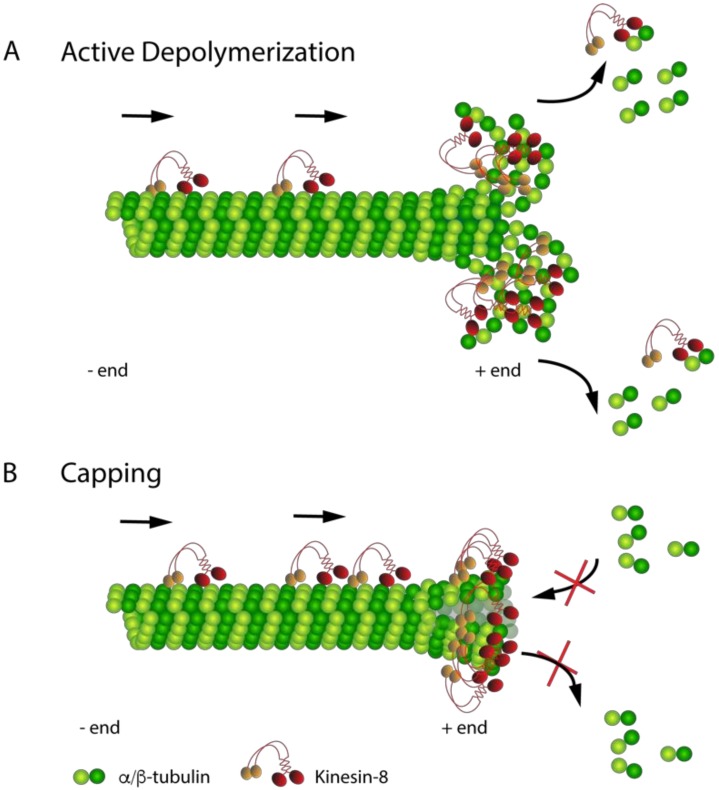
Proposed models for Kinesin-8 MT) depolymerization: (**A**) The active depolymerization model proposes that the Kinesin-8 motor protein walks on the MT lattice towards the plus end where it dwells for some time before depolymerizing the MT. (**B**) The capping model proposes that the Kinesin-8 protein, particularly human Kif18A, suppresses MT dynamics at the plus ends by serving as a capping factor. Capping blocks both growth and shrinkage, ultimately leading to MT catastrophe and depolymerization of the MT lattice.

**Figure 2 biomolecules-09-00001-f002:**
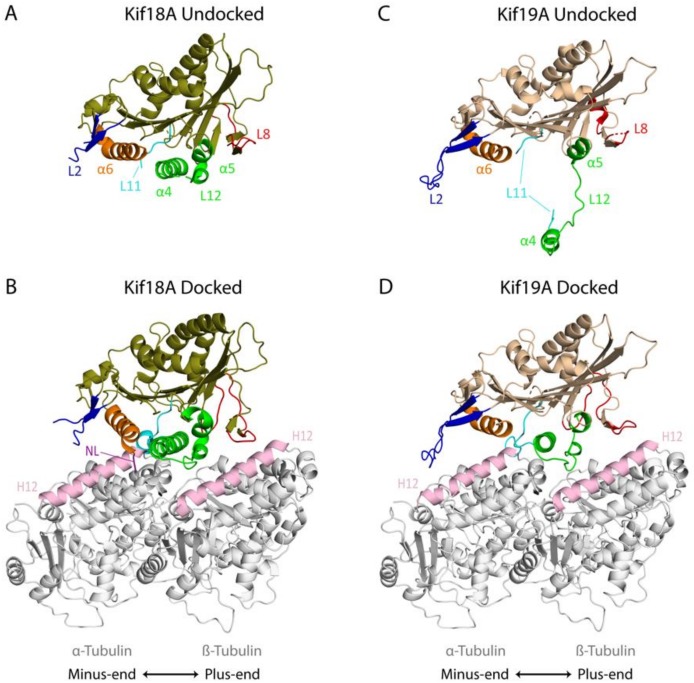
Crystal structures of Kinesin-8 proteins and their cryo-EM MT docked state: (**A**) Undocked Kif18A complexed with Mg^2+^-ADP (PDB: 3LRE) [82]. (**B**) Nucleotide free Kif18A docked to the MT (PDB: 5OAM) [77]. (**C**) Undocked Kif19A complexed with Mg^2+^-ADP (PDB: 5GSZ) [79]. (**D**) Nucleotide free Kif19A docked to the MT (PDB: 5GSY; tubulin coordinates provided by N. Hirokawa) [79]. Key elements required for MT binding and/or MT destabilizing activity are color coded: Switch II cluster (α4-L12-α5) is shown in green, loop L2 in blue, helix α6 in orange, loop L11 in cyan, loop L8 in red, and neck linker (NL) in purple. Helix 12 of α- and β-tubulin is shown in light pink.

**Figure 3 biomolecules-09-00001-f003:**
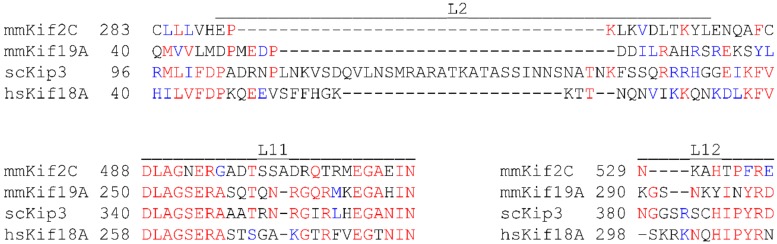
Sequence alignment of critical elements of the motor domain associated with MT depolymerization for the Kinesin-13 (Kif2C) and for the Kinesin-8 (Kif19A, Kip3, and Kif18A) proteins. Identical residues are shown in red, similar residues are shown in blue, and dissimilar residues are shown in black. β-sheet residues on either side of loop L2 are included. Clustal W was used for sequence alignment. Accession numbers: mmKif2C NP_608301.3; mmKif19A NP_001096085.1; scKip3 KZV11013.1; hsKif18A NP_112494.3.

**Table 1 biomolecules-09-00001-t001:** Kinesin-8 family members in different species.

Organism	Family Member(s)
*Aspergillus nidulans*	KipB
*Caenorhabditis elegans*	KLP-13
*Drosophila melanogaster*	KLP67A, Kif19A
*Homo sapiens*	Kif18A, Kif18B, Kif19
*Mus musculus*	Kif18A, Kif18B, Kif19A
*Saccharomyces cerevisiae*	Kip3
*Schizosaccharomyces pombe*	Klp5/Klp6 (heterodimer)
*Xenopus laevis*	Kif18A, Kif18B

**Table 2 biomolecules-09-00001-t002:** Biophysical Properties of Kinesin-8 Proteins.

	Kip3	Klp5/6	Kif18A	hsKif18B	mKif19A	Klp67A
Velocity (µm min^−1^)	3 [31] 0.71 ± 0.23 [33] 3.2 ± 0.3 [72]	*2.3 ± 0.2* [73] *Klp5: 0.4 ± 0.22 ** *Klp6: 5.22 ± 1.08 ** [74]	11.9 ± 2.3 [68] 18.6 ± 5.4 [70]	3.12 ± 0.18 [71] 20.9 ± 0.4 [45]	*1.3 ± 0.18* [37]	*3 ± 1.2* [75]
Run Length (µm)	12.4 ± 2.3 [31] 11 ± 2 [72]	7.2 ± 5.9 [73]	9.4 ± 5.7 [68] 10.1 ± 4.6 [70]	0.74 ± 0.22 [71] >7 [45]	N.D.	N.D.
End Dwell Time (s)	36 ± 4 [72] 38.2 ± 6 [76]	Klp6: 42 ± 24* [74]	~55 [69]	1.42 ± 0.57 [71] 22.8 [45]	N.D.	N.D.
Depolymerization Rate (µm min^−1^)	≤2 [31] 0.06 [33] 2.5–4 [72]	N.S. [73,74]	0.052 ± 0.026 ** [77] N.S. [78] 0.21 ± 0.08–1.25 ± 0.14 [32]	N.S. [45]	1.07 ± 0.23 [37]	N.D.

Velocities represent single molecule velocities, except for those in italics, which were obtained via gliding assays. Values are for full length constructs unless otherwise noted. * Represents tailless homodimers. ** Represents motor domain and neck linker constructs. Not significant (N.S.). Not determined (N.D.). Some values were converted from their original units for ease of comparison.

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
