# Peer review of "Emerging Insights into the Function of Kinesin-8 Proteins in Microtubule Length Regulation"

_biomolecules, 2018, doi:10.3390/biom9010001_

Round 1

Reviewer 1 Report

Shresta et al present a succinct but comprehensive account of recent developments in the understanding of molecular mechanisms of the Kinesin-8 family of microtubule motors. The text provides important background about microtubule dynamics before summarizing cellular localization and roles of this motor family. The majority of review focuses on what is known about the molecular properties of kinesin-8s, and integrates a range of data from both structural and biophysical studies. The text is fair, well-balanced and highlights several open questions remaining in the field. It is clearly written and well-organized, with nicely laid-out figures that are well integrated into the text. I have only a few very minor suggestions for edits:

- p2, lines 43-51: two models of microtubule dynamic instability are described and are presented as being distinct, but they are not necessarily mutually exclusive – this could be clarified;

- p3, line87: to avoid the ambiguous implication that not all Kinesin-8s bind to microtubules, I suggest the text here be reworded “…. not all Kinesin-8 proteins are localized via movement on microtubules, as….” or similar;

- p4, line134: typo “deploymerize”

- p5-6, line 189-190: helix alpha4 of the Kinesin-8 motor domain is described as “the key MT binding domain”, but the use of “domain” in this context is not appropriate for a single secondary structural element; the concept of a domain within a motor domain will also confuse some readers; I suggest “element” here instead, as the authors also use on p7, line 227;

Author Response

Reviewer 1

Shresta et al present a succinct but comprehensive account of recent developments in the understanding of molecular mechanisms of the Kinesin-8 family of microtubule motors. The text provides important background about microtubule dynamics before summarizing cellular localization and roles of this motor family. The majority of review focuses on what is known about the molecular properties of kinesin-8s, and integrates a range of data from both structural and biophysical studies. The text is fair, well-balanced and highlights several open questions remaining in the field. It is clearly written and well-organized, with nicely laid-out figures that are well integrated into the text. I have only a few very minor suggestions for edits:

We thank this reviewer for the careful review of the manuscript and the suggestions to improve clarity.  All changes have been made as outlined below. 

- p2, lines 43-51: two models of microtubule dynamic instability are described and are presented as being distinct, but they are not necessarily mutually exclusive – this could be clarified;

We added the term- “not mutually exclusive” to clarify this point.

- p3, line87: to avoid the ambiguous implication that not all Kinesin-8s bind to microtubules, I suggest the text here be reworded “…. not all Kinesin-8 proteins are localized via movement on microtubules, as….” or similar;

The change has been made.

- p4, line134: typo “deploymerize”

Thank you- typo is corrected.

- p5-6, line 189-190: helix alpha4 of the Kinesin-8 motor domain is described as “the key MT binding domain”, but the use of “domain” in this context is not appropriate for a single secondary structural element; the concept of a domain within a motor domain will also confuse some readers; I suggest “element” here instead, as the authors also use on p7, line 227;

This has been corrected. 

Reviewer 2 Report

The manuscript by Shrestha et al. is a well-written and comprehensive review of Kinesin-8 function with a particular focus on the biochemical mechanisms of microtubule destabilization and length regulation. This is a timely and up-to-date synthesis of the state of our understanding of an important class of microtubule motor and regulatory protein. As such, it will be significant and useful to a wide audience, e.g. cytoskeletal function, mitosis, microtubule regulation, kinesin motor proteins. I believe that the current version of the manuscript is essentially suitable for publication, and I suggest the following minor changes that may improve the manuscript for a general audience.

1) In subheading 1 and/or the beginning of section 1 (lines 19-21) it would be useful to define what MT stands for (i.e. other than in abstract only).

2) A general reader may benefit from a clearer description of MT polarity, i.e. minus and plus ends. At the least consider putting “minus end” and “plus end” in quotes on Line 24-25 to indicate they are designations for the two ends. This will be relevant for latter sections.

3) The sentence ending on Line 93 appears to end with a typo, “orientation and positioning of the metaphase spindle length”. It seems as if ‘length’ should be removed from the end.

4) The sentence on line 120 appears to make clear that not all Kin-8 are highly processive. But then goes on to mention 2 conflicting studies in which Kif18B was found to be not processive or processive. Then it is proposed that future work will be required to determine the differences (i.e. not settled yet). It seems more accurate for the sentence on Line 120 to acknowledge this, e.g. “It is not yet clear whether all Kinesin-8 proteins are highly processive…”.

5) Line 117: For a general reader, would it be helpful to indicate the role of processivity for kinesins, and why/how this may be relevant for Kinesin-8?

6) The sentence on Lines 133-135 appears to have unclear wording near the end. As it reads now, it appears to either have the reference to plus ends erroneously repeated or is trying to say that there is a gradient of Kip3 that increases towards plus ends, but the gradient itself exists only at plus ends. If the latter is correct, it is difficult to discern. Consider revising.

7) In Table 2, for Kip3 velocities 73.8 nm/s may have been obtained via gliding assays (ref 33). If so, it should be in orange text.

8) In the empty boxes of Table 2 it may be clearer to indicate something like N.D. (not determined) rather than leave blank.

9) Line 166, 4th word “reveal” may be too strong to use in this case. It seems “indicate” or “suggest” would be more accurate.

10) The end of the sentence on Line 264 may benefit from including the word ‘arriving’; e.g. “reduction of the amount of motor arriving at MT ends.”

11) In Figure 3 it may be helpful to in some way demark that Kif2c is an example of Kin-13 while the others are Kin-8. In the current form it may be difficult for general readers to discern this from the table.

12) The statement that begins on Line 293 may also be too strongly worded considering the available evidence. For instance the Loop elements that have been removed from Kin-8 decrease depolymerization, but also motility. As the authors convey, the decrease in depolymerization may result from the reduced motility. Yet, the loop may also influence both activities (together or independently). Consider modifying to not sound so absolute.

13) The sentence ending on Line 308 is somewhat unclear to first-time readers. Specifically, it is not clear whether the sentence is indicating that ATP hydrolysis is not required for the Kin-13 mechanism, or the Kin-8 mechanism. Although this is cleared up in the next few sentences, it leaves new readers momentarily uncertain and interrupts the flow of information.

14) The thought conveyed by the last sentence of the first paragraph on page 9 (Lines 318-323) is not accurate. The paragraph presents data that Kin-13 and Kin-8 depolymerize MTs by different mechanisms. These data support two models for Kin-13 and Kin-8 depolymerization mechanisms. This last sentence then states that the fact that the two proposed models are different further supports the idea that the two kinesins utilize different mechanisms. This is circular reasoning, i.e. the conclusions from the data are not further evidence supporting the conclusions.

15) Line 390, this sentence would be clearer if it specified the MTs. For instance, “On antiparallel MTs, the Kip3 motor walks toward the plus end of one MT while the tail remains relatively stationary on the other, resulting in relative sliding of the MTs”.

16) The end of the sentence on Line 393 is unclear. It ends with “due to binding of motors with opposite orientations.” Although it only makes sense in the context of kinesins to be bound in one orientation along the MT, one can imagine orientation referring to minus-to-plus end. Perhaps “due to binding of motors with opposite head-to-tail orientations” would clarify.

17) In the sentence on Line 396 the last word would be more accurately “sliding” activity rather than “crosslinking” activity, as the sliding is affecting spindle length. Consider changing to “sliding” or “crosslinking/sliding”.

Author Response

The manuscript by Shrestha et al. is a well-written and comprehensive review of Kinesin-8 function with a particular focus on the biochemical mechanisms of microtubule destabilization and length regulation. This is a timely and up-to-date synthesis of the state of our understanding of an important class of microtubule motor and regulatory protein. As such, it will be significant and useful to a wide audience, e.g. cytoskeletal function, mitosis, microtubule regulation, kinesin motor proteins. I believe that the current version of the manuscript is essentially suitable for publication, and I suggest the following minor changes that may improve the manuscript for a general audience.

We thank this reviewer for such a careful and thoughtful critique of the manuscript.  The reviewer has done an outstanding job of identifying subtle points where our writing lacked clarity.  The suggested changes are very helpful for the reader and have significantly improved the clarity of the review. 

1) In subheading 1 and/or the beginning of section 1 (lines 19-21) it would be useful to define what MT stands for (i.e. other than in abstract only).

 This has been corrected

2) A general reader may benefit from a clearer description of MT polarity, i.e. minus and plus ends. At the least consider putting “minus end” and “plus end” in quotes on Line 24-25 to indicate they are designations for the two ends. This will be relevant for latter sections.

 We used the quotation suggestion, as we do not yet discuss MT dynamics at this point and did not want to add complexity to the definition of MT structure.

3) The sentence ending on Line 93 appears to end with a typo, “orientation and positioning of the metaphase spindle length”. It seems as if ‘length’ should be removed from the end.

 Thank you- this has been corrected. 

4) The sentence on line 120 appears to make clear that not all Kin-8 are highly processive. But then goes on to mention 2 conflicting studies in which Kif18B was found to be not processive or processive. Then it is proposed that future work will be required to determine the differences (i.e. not settled yet). It seems more accurate for the sentence on Line 120 to acknowledge this, e.g. “It is not yet clear whether all Kinesin-8 proteins are highly processive…”.

 We changed the initial statement to “most”. The proposed future work was meant to reconcile the Kif18B studies and the overall role of processivity. The text has been rewritten for clarity (lines 121-132). 

5) Line 117: For a general reader, would it be helpful to indicate the role of processivity for kinesins, and why/how this may be relevant for Kinesin-8?

 We added a sentence (lines 123-125) to address this point.

6) The sentence on Lines 133-135 appears to have unclear wording near the end. As it reads now, it appears to either have the reference to plus ends erroneously repeated or is trying to say that there is a gradient of Kip3 that increases towards plus ends, but the gradient itself exists only at plus ends. If the latter is correct, it is difficult to discern. Consider revising.

 Excellent point.  We rewrote the sentence to make the main point more clear and to link it back to the processivity experiments in the previous paragraph (lines 154-155).

7) In Table 2, for Kip3 velocities 73.8 nm/s may have been obtained via gliding assays (ref 33). If so, it should be in orange text.

Thank you for catching this.  This prompted us to have one of the authors triple check every value in the table.  We also converted the all values to µm so that the units would be consistent.

8) In the empty boxes of Table 2 it may be clearer to indicate something like N.D. (not determined) rather than leave blank.

 We have made this change. 

9) Line 166, 4th word “reveal” may be too strong to use in this case. It seems “indicate” or “suggest” would be more accurate.

 We have changed to suggest (line 242)

10) The end of the sentence on Line 264 may benefit from including the word ‘arriving’; e.g. “reduction of the amount of motor arriving at MT ends.”

 Good suggestion- we have made this change.

11) In Figure 3 it may be helpful to in some way demark that Kif2c is an example of Kin-13 while the others are Kin-8. In the current form it may be difficult for general readers to discern this from the table.

 We have added this information in the legend.

12) The statement that begins on Line 293 may also be too strongly worded considering the available evidence. For instance the Loop elements that have been removed from Kin-8 decrease depolymerization, but also motility. As the authors convey, the decrease in depolymerization may result from the reduced motility. Yet, the loop may also influence both activities (together or independently). Consider modifying to not sound so absolute.

 We have softened the conclusions (paragraph on lines 372-377)

13) The sentence ending on Line 308 is somewhat unclear to first-time readers. Specifically, it is not clear whether the sentence is indicating that ATP hydrolysis is not required for the Kin-13 mechanism, or the Kin-8 mechanism. Although this is cleared up in the next few sentences, it leaves new readers momentarily uncertain and interrupts the flow of information.

 We have removed the comparison to the Kinesin-13 proteins, which we agree with the reviewer that the sentence was confusing as written. 

14) The thought conveyed by the last sentence of the first paragraph on page 9 (Lines 318-323) is not accurate. The paragraph presents data that Kin-13 and Kin-8 depolymerize MTs by different mechanisms. These data support two models for Kin-13 and Kin-8 depolymerization mechanisms. This last sentence then states that the fact that the two proposed models are different further supports the idea that the two kinesins utilize different mechanisms. This is circular reasoning, i.e. the conclusions from the data are not further evidence supporting the conclusions.

 Excellent point.  We removed the part of the first sentence from “reinforcing the idea” and just present that as data.  In the last sentence we removed “further” supporting so that there is just one conclusion from the data. 

15) Line 390, this sentence would be clearer if it specified the MTs. For instance, “On antiparallel MTs, the Kip3 motor walks toward the plus end of one MT while the tail remains relatively stationary on the other, resulting in relative sliding of the MTs”.

 We made the suggested change.

16) The end of the sentence on Line 393 is unclear. It ends with “due to binding of motors with opposite orientations.” Although it only makes sense in the context of kinesins to be bound in one orientation along the MT, one can imagine orientation referring to minus-to-plus end. Perhaps “due to binding of motors with opposite head-to-tail orientations” would clarify.

 We made the suggested change.

17) In the sentence on Line 396 the last word would be more accurately “sliding” activity rather than “crosslinking” activity, as the sliding is affecting spindle length. Consider changing to “sliding” or “crosslinking/sliding”.

We made the suggested change.